# FeF_3_/(Acetylene Black and Multi-Walled Carbon Nanotube) Composite for Cathode Active Material of Thermal Battery through Formation of Conductive Network Channels

**DOI:** 10.3390/nano13202783

**Published:** 2023-10-17

**Authors:** Su Hyeong Kim, Ji-Hyeok Choi, So Hyun Park, Tae Young Ahn, Hae-Won Cheong, Young Soo Yoon

**Affiliations:** 1Department of Materials Science & Engineering, Gachon University, Seongnam 13120, Republic of Korea; kkd1573@gachon.ac.kr (S.H.K.); joshua9456@gachon.ac.kr (J.-H.C.); psh6575@gachon.ac.kr (S.H.P.); 2Agency for Defense Development (ADD), Daejeon 34186, Republic of Korea; ahnty@add.re.kr

**Keywords:** thermal battery, FeF_3_, cathode active material, multi-walled carbon nanotube, acetylene black

## Abstract

Considerable research is being conducted on the use of FeF_3_ as a cathode replacement for FeS_2_ in thermal batteries. However, FeF_3_ alone is inefficient as a cathode active material because of its low electrical conductivity due to its wide bandgap (5.96 eV). Herein, acetylene black and multi-walled carbon nanotubes (MWCNTs) were combined with FeF_3_, and the ratio was optimized. When acetylene black and MWCNTs were added separately to FeF_3_, the electrical conductivity increased, but the mechanical strength decreased. When acetylene black and MWCNTs were both added to FeF_3_, the FeF_3_/M_1_AB_4_ sample (with 1 wt.% MWCNTs and 4% AB) afforded a discharge capacity of approximately 74% of the theoretical capacity (712 mAh/g) of FeF_3_. Considering the electrical conductivity and mechanical strength, this composition was confirmed to be the most suitable.

## 1. Introduction

A thermal battery is a reserve-type battery that uses molten salt as an electrolyte and is activated at high temperatures (generally 350–550 °C) by heating when its use as a primary battery is required [1,2,3]. Thermal batteries can be stored for a long time; therefore, they are used as a power source for guided missiles in the military or as a power source for hydraulic systems in aircraft for emergency purposes [4,5]. Thermal batteries have developed rapidly in recent years, aiming to ensure stability and reliability when used as an energy storage device in extreme environments (vibration, shock, acceleration, and pressure). The main physical and chemical characteristics required for the active cathode materials of thermal batteries are high thermal stability, high electronic conductivity, low solubility in molten electrolytes, low equivalent weight, good discharge kinetics, and reasonable cost [6,7].

Simply stated, the electrolyte in a thermal battery exists in a salt state with low reactivity at room temperature, which blocks contact between the anode and cathode. In this state, the movement of Li^+^ ions is controlled, and the self-discharge is kept low, enabling long-term storage. When the use of the battery is required, the user supplies heat by activating an ignitor, which liquefies the electrolyte and supplies power through an electrochemical reaction. The most commonly used thermal batteries employ Li-Si as the anode, iron disulfide (FeS_2_) as the cathode, and LICl-KCl (or three components) as the electrolyte. FeS_2_ has the advantages of high conductivity in batteries, low cost, and a high theoretical capacity [8,9,10]. The reaction equations for a thermal battery using Li-Si as the anode and FeS_2_ as the cathode are as follows [11]:(1)$$4Li+3FeS_{2}\to {Li}_{4}{Fe}_{2}S_{5}+FeS\;\left(2.1\;V\right)$$
(2)$$2Li+{Li}_{4}{Fe}_{2}S_{5}+FeS\to 3{Li}_{2}FeS_{2}\;\left(1.9\;V\right)$$
(3)$$6Li+3{Li}_{2}FeS_{2}\to 6{Li}_{2}S+3Fe\;\left(1.6\;V\right)$$

Among these materials, FeS_2_, a core material, decomposes into Fe_7_S_8_ or S_2_ in the operating environment of thermal batteries, i.e., 550 °C, lowering the discharge capacity [12,13]. The sulfur gas formed in the decomposition process seriously impacts the battery cathode and battery components and carries the risk of short circuits or battery explosions, raising safety issues [14].

Military equipment has become smarter with the recent development of cutting-edge defense technologies. Advancements have increased the precision and range of conventional artillery shells. Currently, a high-output power source that can overcome the spatial constraints of applications such as missiles and ensure miniaturization and precision is required. Therefore, there is an increasing need to develop new active cathode materials to solve the problems associated with FeS_2_ in thermal batteries.

Currently, research on chlorides (NiCl_2_) [15,16,17], fluorides (FeF_3_, NiF_2_) [18,19], and oxides as next-generation active cathode materials for thermal batteries is in progress [20,21]. Among them, transition metal fluorides (TMFs) have attracted significant interest in academia [22]. TMFs have a high theoretical capacity owing to their multielectron conversion mechanism [23].
(4)$$MFx+x{Li}^{+}+xe^{-}↔xLiF+M$$

The reaction of iron fluoride (FeF_3_), a type of TMF, proceeds via a two-step lithium (Li) intercalation and conversion reaction, affording a high theoretical capacity of 712 mAh/g.
(5)$$FeF_{3}+Li\to LiFeF_{3}\;\left(4.5–2.5\;V\right)$$
(6)$$LiFeF_{3}+Li\to 3LiF+Fe\;(under\;2.5\;V)$$

The Li intercalation reaction proceeds between 4.5 V and 2.5 V, and the conversion reaction proceeds below 2.5 V [24]. FeF_3_ has a theoretical capacity of 712 mAh/g, which is lower than that of FeS_2_, but has an energy density of 1947 Wh/kg, which is higher than that of FeS_2_ (1273 Wh/kg) [25]. Additionally, FeF_3_ is a candidate as a non-sulfide-based cathode active material due to its high-temperature stability above 550 °C. However, FeF_3_ has low electrical conductivity because of its wide bandgap (5.96 eV), and LiF, the product of the conversion reaction, is an insulator [26,27]. Therefore, the use of FeF_3_ alone as an active cathode material in thermal batteries is inefficient.

In this study, multi-walled carbon nanotubes (MWCNTs), a carbon material widely used as a conductive material, and acetylene black (AB) were combined with FeF_3_ in a composite to overcome the conductivity limitations of FeF_3_. The respective composites of AB and MWCNTs with FeF_3_ exhibit improved electrical conductivity, but the durability of the pelletized FeF_3_ composite was relatively low owing to the spring-back effect and agglomeration of the carbon materials. Spring-back is the geometric change made to a part at the end of the forming process when the part has been released from the forces of the forming tool. In the case of MWCNTs, this is an effect in which MWCNTs, pressed due to the pressure applied during the pelletizing process, try to return to their original state after the pressure is released. To solve this problem, a composite was fabricated by mixing AB and MWCNTs with FeF_3_ in an appropriate ratio. When the FeF_3_ composite was fabricated using only MWCNTs, the MWCNTs were not located between the FeF_3_ particles. Therefore, a uniform conductive network could not be formed; the same was true for AB. When an FeF_3_ composite was produced with AB and MWCNTs, the possibility of forming a conductive network between all FeF_3_ particles increased, and the durability of the pelletized FeF_3_ composite could be secured. The details are shown in Figure 1. The goal was to develop an organic binder-free cathode active material that overcame the low electrical conductivity of FeF_3_ for use as a high-density, high-voltage cathode active material in thermal batteries.

## 2. Materials and Methods

### 2.1. Reagents

Iron fluoride (FeF_3_, Sigma-Aldrich, St. Louis, MO, USA), iron disulfide (FeS_2_, Alpha-Aesar, Haverhill, MA, USA), multiwalled carbon nanotubes (COOH-functionalized MWCNTs, US Research Nanomaterials, Houston, TX, USA), acetylene black (Graphene Supermarket, Ronkonkoma, NY, USA), and absolute ethanol (SAMCHUN, Pyungtaek, Republic of Korea) were used without additional purification.

### 2.2. Experimental Details

To fabricate the FeF_3_ composite, FeF_3_, AB, and the MWCNTs were weighed in a ball-milling pot according to their respective weight ratios. The mixture was subjected to low-speed ball milling using 10 mm and 5 mm zirconia balls (powder: zirconia ball = 1:7 wt%). The ball-milled FeF_3_ composite was used as a slurry for coin cell fabrication. Additional samples were dried and pelletized to measure the resistance, electrical conductivity, and mechanical strength. All experiments were conducted in a dry laboratory at 20% humidity, and the coin cells were assembled in a glove box under an argon environment. Owing to the difficulties in measuring the cathode discharge capacity in a thermal battery environment, the discharge capacity was measured in the laboratory using a half-cell with a Li metal (anode) and LiPF_6_ (electrolyte).

### 2.3. Characterization

The effectiveness of the acid treatment of the MWCNTs was confirmed by Raman spectroscopy (Olympus, Tokyo, Japan, BX51), and the phase change of FeF_3_ was identified by X-ray diffraction (XRD, Rigaku, SmartLab, Eagle Farm, Australia) after low-speed ball milling. The morphology of the samples was observed using a scanning electron microscope (SEM, Hitachi, Tokyo, Japan, SU8600), and the distribution of the elements was confirmed by energy-dispersive X-ray spectroscopy (EDS) analysis. Through thermogravimetric analysis (TGA, TA Instruments, New Castle, DE, USA, SDT Q600), the high-temperature stability was determined from the weight change versus the temperature profile of each sample, and the activation energy was calculated. The TGA was analyzed from room temperature to 800 °C, at a temperature increase rate of 10 °C/min, and in the N_2_ atmosphere. The battery discharge capacity was analyzed using a battery cycler (WonATech, Seoul, Republic of Korea, WBCS30000S). The discharge capacity was measured at a c-rate of 0.1 C, and the cutoff voltage was set at 0.5 V. The sheet resistance and specific resistance were measured using a four-point probe (Keithley, Cleveland, OH, USA, 2450 source meter) [28] and multi-meter probe (Fluke, Everett, DC, USA, 287 True RMS Multimeter), and the mechanical strength was measured using a compressive strength meter (ZwickRoell, Ulm, Germany, Z100SN). The sheet resistance of the pelletized active material was measured with a four-point probe, and the resistance in the horizontal and vertical directions of the pellet was measured with a multi-meter probe to confirm the tendency of the two results. The electrical conductivity was calculated using resistance measured with a four-point probe. The mechanical strength was measured using a 3 N load cell, and the maximum load was measured by setting the pellet vertically on the lower jig and lowering the upper jig at a speed of 5 mm/min. The pellet used at this time was manufactured with a diameter of 12 mm by applying a pressure of 30 kN.

## 3. Results

Figure 2a shows the TGA data for bare FeF_3_ and FeS_2_. FeS_2_ was stable, showing a weight loss rate of approximately 2.5% up to 450 °C, followed by rapid decomposition, with a weight loss rate of approximately 25% at 700 °C. On the other hand, FeF_3_ continued to lose weight even when the weight loss due to moisture (up to 250 °C) was excluded. However, there was no rapid decrease in the weight loss with increasing temperature, where the weight loss rate at 700 °C was ~10%, indicating better thermal stability compared with FeS_2_. Because the thermal decomposition of the cathode active material in a thermal battery operating environment was a critical drawback affecting the battery performance, FeF_3_ was selected instead of FeS_2_.

Figure 2b shows the Raman spectra of the MWCNTs. The “D” peak reflects disorder in the MWCNTs, and the “G” peak is characteristic of carbon materials. The ratio of the intensity of the D peak to that of the G peak indicates the defect rate of the material. The *I_D_/I_G_* value of the MWCNTs was 1.02, and the *I_D_/I_G_* of the acid-treated MWCNTs was 1.45. The defect rate increased as the value of *I_D_/I_G_* increased; thus, the acid-treated MWCNTs had more defects. The acid-treated MWCNTs were more easily dispersed through the low-speed ball-milling process for producing a slurry of the samples for fabricating coin cells. In the COOH-functionalized MWCNTs, the π-conjugation was disrupted, and a surface dipole moment was introduced, which increased the dispersibility in solution [29,30]. To prepare the FeF_3_/MWCNT composite, ethanol was added during mechanical dispersion via low-speed ball milling. Increasing the dispersibility of the MWCNTs was an important factor for the fabrication of the FeF_3_/MWCNT composites. Therefore, the acid-treated MWCNTs were used in this study.

Figure 3a shows the XRD data after mixing 20 wt% AB and MWCNTs, respectively, with FeF_3_ via low-speed ball milling. Hereafter, the samples mixed with AB are referred to as FeF_3_/AB, and the samples mixed with MWCNTs are termed FeF_3_/M. Because the sample fabrication process involves mechanical mixing using low-speed ball milling, it is possible that the phase of FeF_3_ may change. However, a comparison of the XRD pattern with ICDD 00-033-0647 confirmed that the FeF_3_ phase was maintained and that the sample used in the experiment had the characteristics of FeF_3_. Figure 3b,c show the SEM and EDS images after mixing 20 wt% AB and MWCNTs, respectively, with FeF_3_ via low-speed ball milling. FeF_3_ initially had a rhombohedral shape but adopted a spherical shape after the ball-milling process [31] As shown in Figure 3b, the particles of both AB and FeF_3_ were spherical, although FeF_3_ and AB could be distinguished via EDS mapping, which confirmed that AB was distributed throughout the composite. Figure 3c shows the composite of FeF_3_ and the MWCNTs, where tubular MWCNTs were observed. Similar to AB, the MWCNTS were distributed throughout the composite, as confirmed through EDS mapping.

Figure 4 shows the TGA data after mixing FeF_3_ with up to 20 wt% AB and MWCNTs, with low-speed ball milling. The FeF_3_/AB composites with 2.5, 5.0, 10.0, and 20.0 wt% AB showed 17%, 19%, 26%, and 25% weight loss up to 700 °C. The composites with 2.5, 5.0, 10.0, and 20.0 wt% MWCNTs showed weight loss rates of 8.7%, 10.8%, 13%, and 12.3%, respectively. The composites with AB underwent rapid weight loss after 400 °C, whereas the composites with the MWCNTs underwent a gradual weight loss after 550 °C. The weight loss before 250 °C was due to moisture evaporation, and the weight loss after 250 °C was due to the sublimation of AB and the MWCNTs. The composites with 10 wt% and 20 wt% AB showed a clear weight loss compared with the samples without AB. Because AB was thermally decomposed around 400 °C [32], the composites containing a greater amount of AB underwent greater weight loss. However, for all four composites with the MWCNTs, there was no significant difference in the weight loss. This was because the MWCNTs decomposed at a higher temperature than AB, that is, above 550 °C [33].

Table 1 shows the resistance, electrical conductivity, and mechanical strength of the composites, measured with a four-point probe and a multi-meter probe. As the amounts of added AB and MWCNTs increased, the resistance decreased, and the electrical conductivity increased. Unlike conventional batteries, thermal batteries use pelletized electrode materials. Therefore, good formability is required for pelletization. However, AB is agglomerated by van der Waals and electrostatic forces, making complete dispersion through mechanical processes difficult [34], whereas in the case of the MWCNTs, pelletization is difficult when pressure is applied owing to the spring-back effect [35]. This explains why the mechanical strength was not constant, as listed in Table 1. Therefore, the addition of AB and MWCNTs was optimized. The comparison of the electrical conductivity showed that the electrical conductivity was noticeably improved when both AB and the MWCNTs were added at 5 wt%. This was because 2.5 wt% of AB or MWCNT was insufficient to create a conductive network throughout the pellet; therefore, the particles acted as a resistance, whereas it was assumed that 5 wt% of AB or MWCNT was sufficient to create a conductive network throughout the pellet.

Figure 5 shows the discharge capacity data for the composites with AB and MWCNTs in different ratios. The measured discharge capacity for the composites with 2.5, 5.0, 10.0, and 20.0 wt% AB was 414.5, 394.8, 389.4, and 363.1 mAh/g, respectively. The corresponding values for the composites with 2.5, 5.0, 10.0, and 20.0 wt% MWCNTs were 435.4, 452.9, 442.4, and 324.5 mAh/g, respectively. The discharge capacity was similar for all composites with AB, and only the composite with 20 wt% MWCNTs had a low discharge capacity. The composite with 5 wt% AB had a stable discharge profile, and the composite with 5 wt% MWCNTs had the highest discharge capacity. MWCNTs have been widely reported to have a higher electrical conductivity than carbon black (CB) [36,37]. However, as the amount of MWCNTs increased, the charge transfer rate decreased owing to aggregation of the MWCNTs [38]. When AB and MWCNTs were combined with FeF_3_, the electrical conductivity of the composite was improved at the expense of the capacity density because the amount of active material that could be contained in the battery was limited. Therefore, the amount of AB and MWCNTs added to FeF_3_ was optimized. Considering the electrical conductivity, thermal stability, and discharge capacity, it was determined that adding 5 wt% AB or MWCNTs was optimal. The carbon material added to FeF_3_ was fixed at 5 wt%. The AB and MWCNTs were mixed with FeF_3_ in various ratios, and the resulting composites were analyzed.

Figure 6 shows the TGA graph for the composite with a fixed carbon material to FeF_3_ ratio of 5 wt%, where the ratios of the AB and MWCNTs were varied. Hereafter, the samples mixed with AB and MWCNTs at various ratios are denoted as FeF_3_/M(x%)/AB(5 − x%). The weight loss rates up to 700 °C were 19.5%, 21%, 19.5%, and 15.7% for FeF_3_/M(1 wt%)/AB(4 wt%), FeF_3_/M(2 wt%)/AB(3 wt%), FeF_3_/M(3 wt%)/AB(2 wt%), and FeF_3_/M(4 wt%)/AB(1 wt%), respectively. Among the samples, FeF_3_/M(4 wt%)/AB(1 wt%), which had the highest MWCNT content, showed the lowest weight loss up to 700 °C. As explained above, because the MWCNTs had better thermal stability than AB, the sample with the highest MWCNT content had the lowest weight loss rate. The remaining samples exhibited similar weight loss rates without significant changes. However, at 550 °C, the operating temperature of the thermal battery, all samples were thermally stable with a weight loss rate of less than 10%.

Table 2 shows the resistance, electrical conductivity, and mechanical strength data after fixing the carbon material added to FeF_3_ at 5 wt% and mixing the AB and MWCNTs at various ratios. The resistance and electrical conductivity were similar to or higher than those of the samples containing either AB or MWCNTs. The mechanical strength was also greatly improved compared with that of FeF_3_ mixed with only AB or MWCNTs. Since the cathode active material of a thermal battery is used by pelletizing it through thermal compression, the mechanical strength characteristics of the pelletized cathode active material are an important factor. Therefore, presenting mechanical strength serves as a strength when compared with other studies [18]. The MWCNTs were randomly arranged between the FeF_3_ particles, and empty spaces were inevitably formed between them. As AB entered this empty space, the empty space was reduced, and the packing density increased, thereby improving the mechanical strength of the pellet. This was attributed to the structural characteristics of the spherical AB and tube-shaped MWCNTs.

Figure 7 shows the discharge capacity after fixing the carbon material added to FeF_3_ at 5 wt% and introducing AB and MWCNTs at various ratios. When the cutoff voltage was set as 0.5 V, it was confirmed that all samples except the FeF_3_/M_2_AB_3_ sample had a discharge capacity of 500–550 mAh/g, which was 70–77% of the theoretical capacity of FeF_3_ (712 mAh/g). Figure 8 shows the activation energy obtained from the TGA data shown in Figure 6. The formula for calculating the activation energy is as follows:(7)$$log\frac{-log(1-\alpha )}{T^{2}}=log\frac{AR}{\beta E_{a}}\left(1-\frac{2RT}{E_{a}}\right)-\frac{E_{a}}{2.303RT}$$
where α is the fraction of the sample decomposed at time *t*, given by
(8)$$\alpha =\frac{W_{i}-W_{t}}{W_{i}-W_{f}}$$
*β* is the linear heating rate, *T* is the absolute temperature (K), *E_a_* is the activation energy (K), *A* is the frequency factor, *R* is the gas constant 8.314 J/mol·K, *w_i_* is the initial weight, *w_t_* is the weight at a given temperature, and *w_f_* is the final weight after completion of the reaction.

The activation energy was calculated to account for the intercalation and discharge efficiency of Li^+^ ions. The activation energy of FeF_3_/M_1_/AB_4_, FeF_3_/M_2_/AB_3_, FeF_3_/M_3_/AB_2_, and FeF_3_/M_4_/AB_1_ was 20.57, 26.33, 22.95, and 22.88 J/mol, respectively. The low activation energy indicated that the movement path of Li^+^ was short. When the Li^+^ movement distance was short, the Li^+^ intercalation and discharge efficiencies increased [39,40]. The sample with the lowest activation energy was the FeF_3_/M_1_/AB_4_ sample, which also had the highest discharge capacity, apart from the FeF_3_/M_4_/AB_1_ sample. However, the FeF_3_/M_4_/AB_1_ sample was unstable in the low-voltage section. Therefore, considering the electrical conductivity, mechanical strength, discharge capacity, activation energy, and thermal stability, it can be concluded that FeF_3_/M_1_/AB_4_ had the most suitable ratio of AB to MWCNTs. The discharge capacity and speed of a thermal battery depend on the microstructure and shape of the electrode, in addition to the unique physicochemical properties of the electrode material [41,42,43]. Because the capacity of a battery has a complex relationship with its components, the ratio of the FeF_3_ composite in this study could not be considered perfect. However, by giving FeF_3_ electrical conductivity, it showed its potential as a thermal battery cathode active material, and further research is needed to determine whether FeF_3_ can perform better as a thermal battery cathode active material through complex formation with other carbon materials or metal plating. Consequently, this experimental result shows the potential as a cathode active material that can overcome the drawbacks of FeS_2_, which is currently used as a thermal battery cathode active material.

## 4. Conclusions

To use FeF_3_, which has high energy density but low electrical conductivity, as the active material for a thermal battery cathode, composites of FeF_3_ with AB and MWCNTs were prepared. The ratio of the carbon material was varied from 2.5 wt% to 20 wt%, confirming that 5 wt% was the most suitable ratio. The composite of FeF_3_ with both AB and MWCNTs exhibited advantages in terms of the electrical conductivity, mechanical strength, and discharge capacity. The composite afforded a discharge capacity of approximately 74% compared with the theoretical capacity of 712 mAh/g of FeF_3_. Regarding using FeF_3_ as a thermal battery cathode active material, the thermal stability could not be explained by a half cell using LiPF_6_. However, discharge experiments were conducted using FeF_3_ and carbon materials without using an organic binder so that it could be used immediately without additional processes when used in a thermal battery. Additionally, TGA data were used to demonstrate stability in the thermal battery operating environment. Therefore, the FeF_3_/AB/MWCNT composite showed sufficient stability to be used as a cathode active material at high temperatures, which is the operating environment of a thermal battery. Considering the electrical conductivity, mechanical strength, discharge capacity, activation energy, and thermal stability, it was confirmed that FeF_3_/M_1_/AB_4_ (1 wt.% MWCNTs and 4% AB) had the optimal ratio of AB and MWCNTs. The FeF_3_ composite was fabricated by mixing AB and MWCNTs, and the characteristics of the composites with specific ratios were compared. This demonstrated the possibility of using the FeF_3_/M_1_/AB_4_ composite as the active material for a thermal battery cathode.

## Figures and Tables

**Figure 1 nanomaterials-13-02783-f001:**
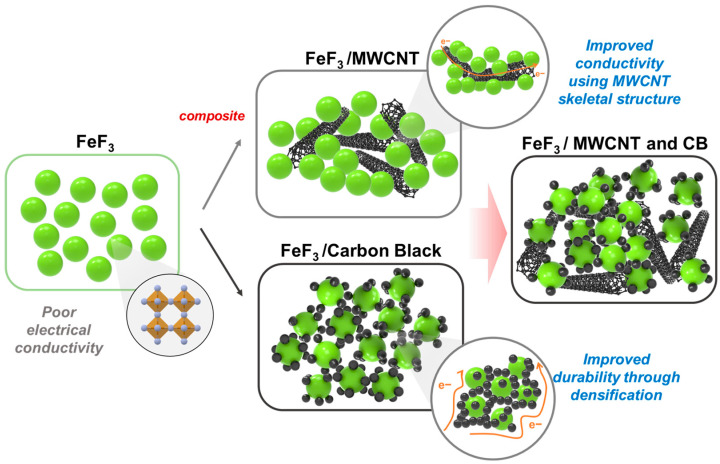
Schematic diagram of the mechanism for improving conductivity and durability through the addition of acetylene black and MWCNT.

**Figure 2 nanomaterials-13-02783-f002:**
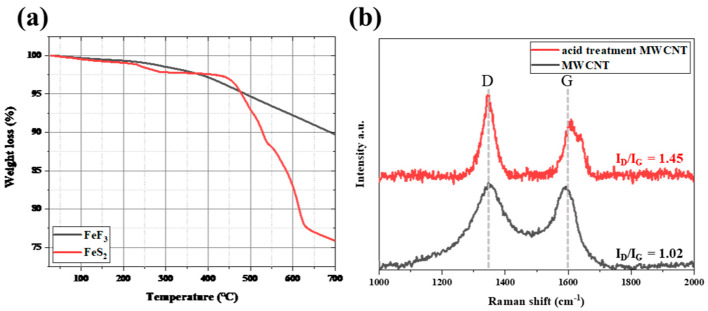
TGA data for FeS_2_ and FeF_3_ (**a**) and Raman spectra of the MWCNTs in the range of 1000–2000 cm^−1^ (**b**).

**Figure 3 nanomaterials-13-02783-f003:**
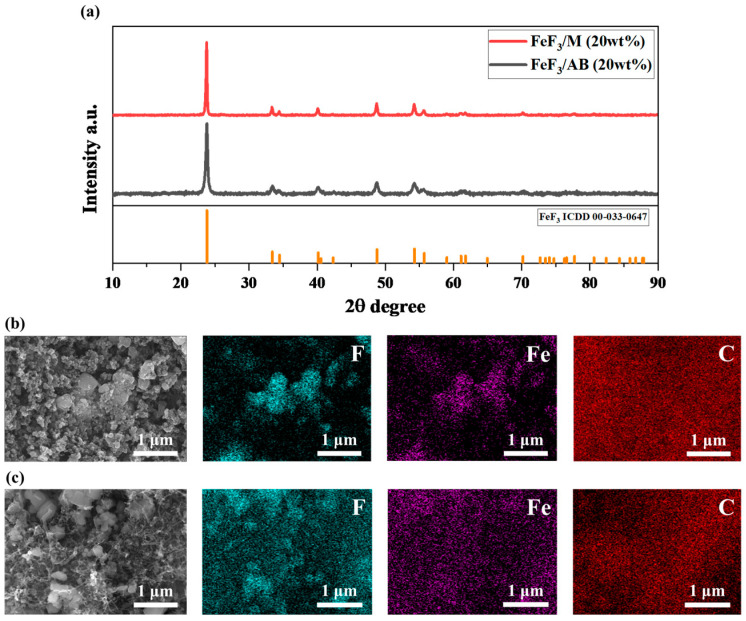
(**a**) XRD data for FeF_3_ composites with 20 wt% AB and MWCNTs. SEM image of FeF_3_ composite with 20 wt% AB (**b**) and MWCNTs (**c**).

**Figure 4 nanomaterials-13-02783-f004:**
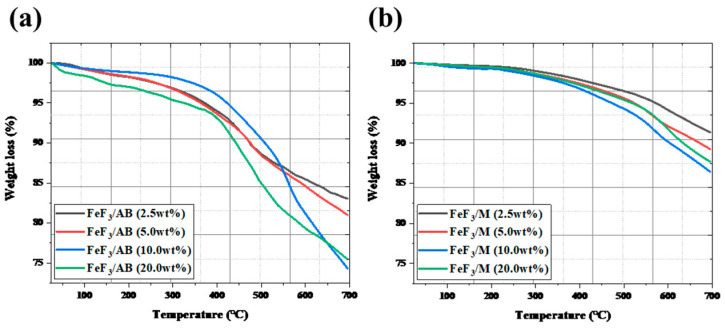
TGA data for FeF_3_ composites of AB (**a**) and MWCNTs (**b**) in various ratios.

**Figure 5 nanomaterials-13-02783-f005:**
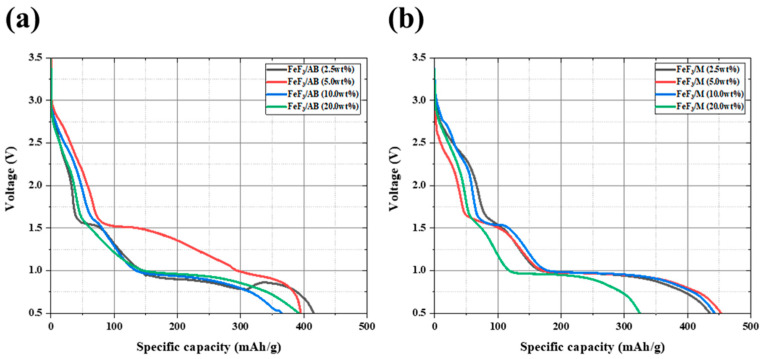
Discharge data for FeF_3_ composite with AB (**a**) and MWCNT (**b**) at various ratios.

**Figure 6 nanomaterials-13-02783-f006:**
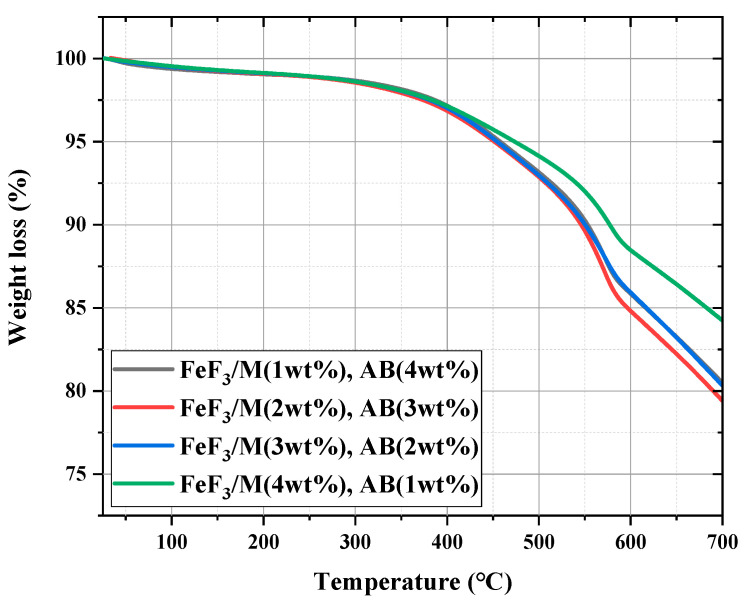
TGA data for FeF_3_ composite with AB and MWCNTs at various ratios.

**Figure 7 nanomaterials-13-02783-f007:**
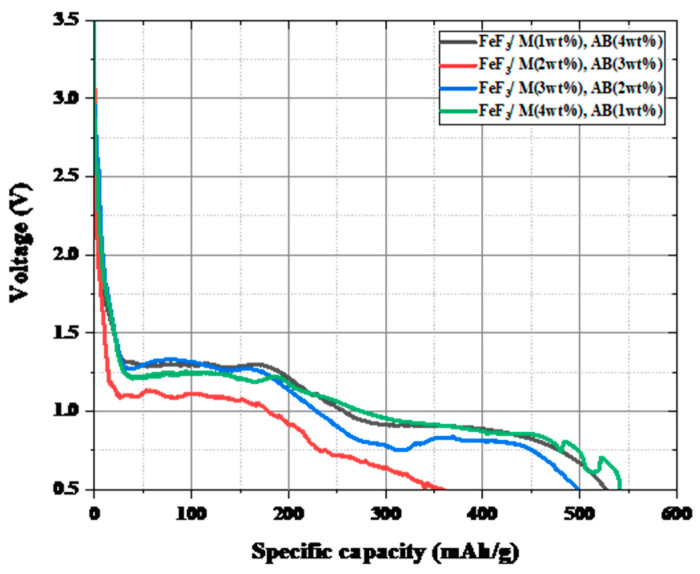
Discharge data for FeF_3_ composite with AB and MWCNTs at different ratios.

**Figure 8 nanomaterials-13-02783-f008:**
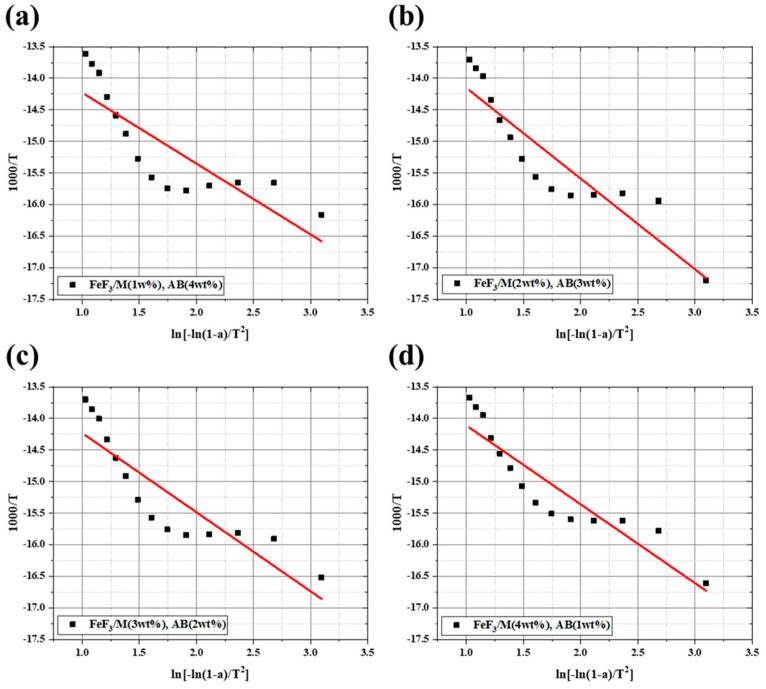
Activation energy of FeF_3_ composite with AB and MWCNTs at different ratios. (**a**) FeF_3_/M_1_/AB_4_; (**b**) FeF_3_/M_2_/AB_3_; (**c**) FeF_3_/M_3_/AB_2_; (**d**) FeF_3_/M_4_/AB_1_.

**Table 1 nanomaterials-13-02783-t001:** Sheet resistance, vertical resistance, electrical conductivity, and mechanical strength of respective FeF_3_ composites.

Type	Sheet Resistance (Ω/sq)	Multi-Meter Probe (Ω/mm)	Electrical Conductivity (S/m)	Mechanical Strength (MPa)
FeF_3_	2.590 × 10^6^	Over-load	4.152 × 10^−4^	0.2968
FeF_3_/AB (2.5 wt%)	2.064 × 10^6^	6091	6.291 × 10^−4^	0.3723
FeF_3_/AB (5.0 wt%)	312.9	2205	4.097	0.3175
FeF_3_/AB (10.0 wt%)	14.78	49.19	76.91	0.2099
FeF_3_/AB (20.0 wt%)	4.756	16.15	182.8	-
FeF_3_/M (2.5 wt%)	1.249 × 10^6^	105.3	8.090 × 10^−4^	0.9007
FeF_3_/M (5.0 wt%)	342.4	27.42	3.281	0.9301
FeF_3_/M (10.0 wt%)	9.460	11.53	113.7	1.4189
FeF_3_/M (20.0 wt%)	6.379	0.78	168.6	0.9975

**Table 2 nanomaterials-13-02783-t002:** Sheet resistance, vertical resistance, electrical conductivity, and mechanical strength of FeF_3_/AB/MWCNT composites.

Type	Sheet Resistance (Ω/sq)	Multi-Meter Probe (Ω/mm)	Electrical Conductivity (S/m)	Mechanical Strength (MPa)
FeF_3_/M_1_AB_4_	140.4	1053	8.281	1.556
FeF_3_/M_2_AB_3_	106.9	3809	10.39	1.182
FeF_3_/M_3_AB_2_	389.8	374.4	2.882	0.8184
FeF_3_/M_4_AB_1_	223.6	537.0	5.081	0.9611

## Data Availability

Not applicable.

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
