# Peer review of "FeF3/(Acetylene Black and Multi-Walled Carbon Nanotube) Composite for Cathode Active Material of Thermal Battery through Formation of Conductive Network Channels"

_nanomaterials, 2023, doi:10.3390/nano13202783_

Round 1

Reviewer 1 Report

In this work, the authors studied the effect of AB and MWCNTs on forming an ideal FeF3 composite for thermal batteries. They found that the composite of FeF3 with both AB and MWCNTs exhibited advantages in terms of the electrical conductivity, mechanical strength, and discharge capacity. The composite afforded a discharge capacity of approximately 74% compared with the theoretical capacity of 712 mAh/g of FeF3. Combined with other tests, the authors found that FeF3/M1/AB4 (1 wt.% MWCNTs and 4% AB) had the optimal ratio of AB and MWCNTs.

Overall, the manuscript is clearly written and some interesting findings are presented by the authors. However, there are still some issues that the authors have to address.

In the main text, the authors state the “spring back effect”. The reviewer wonders what is the spring back effect?

During the experiment, the authors employed a half-cell using Li metal and LiPF6 electrolyte in the laboratory to study the discharge capacity of the cathode capacity. The reviewer wonders if this test can reliably represent a thermal battery operating at high temperature?

Why the authors prepare the acid-treated MWCNTs? Please clarify this in the main text or in the experiment section.

The authors state that 5 wt% of AB or MWCNT was sufficient to create a conductive network throughout the pellet, the reviewer wonders based on what facts did the authors reach this conclusion.

Finally, it seems to the reviewer that there are some discussions in the conclusion part. It is suggested that the authors conducted the discussion before the conclusion part, not within it.

Reviewer 2 Report

This is a paper focusing on the evaluation of the effect of acetylene black on the electric conductivity in order to improve electrochemical performance of FeF3. The authors had performed a series of experiments and calculated the activation energies from the TGA measurements.

The following is a list of comments.

1. 1)     The paper is disorganized with the TGA measurements and the capacities spread over the paper. The authors need to place the TGA measurements together and the capacities together, and then discuss the difference.

2. 2)     The calculations of the activation energies are misleading, since it is impossible to use the Arrhenius relation to fit the data. Also, what is the purpose of calculating the activation energies?

3.  3)    The mechanism for the increase of the electrical conductivity is attributed to the improvement in the conducting paths for electrons. This likely is not the main reason. It is likely due to the increase in the reaction sites for the intercalation of lithium. Had the authors used different sizes of FeF3?

4.  4)    Please provide the environment for the TGA measurements.

5. 5)     How did the authors measure the mechanical strength?

The writing is OK.

Reviewer 3 Report

This paper reports the synthesis and characterization of FeF3/(Acetylene Black and MWCNT) composite as a cathode active material for thermal batteries. The authors claim that the composite has improved electrical conductivity, mechanical strength, and discharge capacity compared to pure FeF3. The paper is well-organized and clearly written, but some issues need to be addressed before publication.

The authors should provide more background information on the motivation and significance of using FeF3 as a cathode material for thermal batteries. How does it compare with other transition metal fluorides or other types of cathodes? What are the challenges and opportunities of using FeF3 in thermal batteries?

The authors should explain the mechanism of how AB and MWCNTs enhance the performance of FeF3. How do they affect the structure, morphology, and electrochemical properties of FeF3? What is the optimal ratio of AB and MWCNTs for achieving the best performance? How does the composite overcome the low electrical conductivity and thermal stability of FeF3?

The authors should provide more experimental details on the fabrication and characterization of the composite. How did they control the ratio and dispersion of AB and MWCNTs in FeF3? How did they measure the resistance, electrical conductivity, mechanical strength, and discharge capacity of the composite?3 What are the error bars and reproducibility of the measurements?

The authors should compare their results with other relevant studies in the literature. How does their composite perform in terms of energy density, power density, voltage, capacity, and cycling stability compared to other cathode materials for thermal batteries? What are the advantages and limitations of their composite?

The abstract should be more concise and informative. It should summarize the main findings and contributions of the paper, not just describe the methods and materials used.

The introduction should provide more references to support the statements and claims made by the authors. Some of the references are outdated or irrelevant to the topic of the paper.

The figures should be improved in terms of quality and clarity. Some of the figures are too small or blurry to read. The labels, legends, and captions should be consistent and descriptive.

The discussion should be more critical and analytical. The authors should discuss the implications and applications of their findings, as well as the limitations and challenges of their work. They should also suggest possible directions for future research.

The conclusion should be more concise and conclusive. It should highlight the main findings and contributions of the paper, not just repeat what has been said in the previous sections.

The formatting and style of the paper should follow the journal’s guidelines. There are some inconsistencies and errors in the citation, punctuation, spelling, grammar, and terminology used in the paper.

Round 2

Reviewer 2 Report

The authors have addressed teh comment to some extent.

OK.